# Genetic Nurture Effects on Type 2 Diabetes Among Chinese Han Adults: A Family-Based Design

**DOI:** 10.3390/biomedicines13010120

**Published:** 2025-01-07

**Authors:** Xiaoyi Li, Zechen Zhou, Yujia Ma, Kexin Ding, Han Xiao, Tao Wu, Dafang Chen, Yiqun Wu

**Affiliations:** 1Department of Epidemiology and Biostatistics, School of Public Health, Peking University, Beijing 100191, China; sherryli@bjmu.edu.cn (X.L.); 13260033986@163.com (Y.M.); 15801298733@163.com (K.D.); gesangmeiduo@pku.edu.cn (H.X.); twu@bjmu.edu.cn (T.W.); 2The Key Laboratory of Geriatrics, Beijing Institute of Geriatrics, Institute of Geriatric Medicine, Chinese Academy of Medical Sciences, Beijing Hospital/National Center of Gerontology of National Health Commission, Beijing 100730, China; peevesomega@163.com; 3Key Laboratory of Epidemiology of Major Diseases (Peking University), Ministry of Education, Beijing 100191, China

**Keywords:** type 2 diabetes mellitus, genetic susceptibility, gene, genetic nurture effect, paternal nurture effect

## Abstract

**Background/Objectives**: Genes and environments were transmitted across generations. Parents’ genetics influence the environments of their offspring; these two modes of inheritance can produce a genetic nurture effect, also known as indirect genetic effects. Such indirect effects may partly account for estimated genetic variance in T2D. However, the well-established specific genetic risk factors about genetic nurture effect for T2D are not fully understood. This study aimed to investigate the genetic nurture effect on type 2 diabetes and reveal the potential underlying mechanism using publicly available data. **Methods**: Whole-genome genotyping data of 881 offspring and/or their parents were collected. We assessed SNP-level, gene-based, and pathway-based associations for different types of genetic effects. **Results**: Rs3805116 (β: 0.54, *p* = 4.39 × 10^−8^) was significant for paternal genetic nurture effects. *MRPS33* (*p* = 1.58 × 10^−6^), *PIH1D2* (*p* = 6.76 × 10^−7^), and *SD1HD* (*p* = 2.67 × 10^−6^) revealed significantly positive paternal genetic nurture effects. Five ontologies were identified as enrichment in both direct and indirect genetic effects, including flavonoid metabolic process and antigen processing and presentation via the MHC class Ib pathway. Two pathways were only enriched in paternal genetic nurture effects, including the transforming growth factor beta pathway. Tissue enrichment of type 2 diabetes-associated genes on different genetic effect types was performed using publicly available gene expression data from the Human Protein Atlas database. We observed significant gene enrichment in paternal genetic nurture effects in the gallbladder, smooth muscle, and adrenal gland tissues. **Conclusions**: *MRPS33*, *PIH1D2*, and *SD1HD* are associated with increased T2D risk through the environment influenced by paternal genotype, suggesting a novel perspective on paternal contributions to the T2D predisposition.

## 1. Introduction

In recent years, the prevalence of type 2 diabetes mellitus (T2D) has rapidly increased; T2D is a major contributor to the global disease burden and to accelerated frailty, disability, hospitalization, institutionalization, and death [1,2,3]. Glycemic control is a primary objective of treatment in the attempt to prevent chronic complications of diabetes. However, poor adherence to self-management behaviors among T2D patients is a major obstacle, leading to low treatment success [4,5]. One possible reason may be the negative nurturing environment [6].

T2D results from multifaceted interactions among genetic, lifestyle, and environmental factors [7]. With respect to genetic risk factors, heritability estimates of T2D range from 47% to 77% in twins of different age groups [8], and previous efforts to use genome-wide association studies (GWASs) have pinpointed 611 genetic susceptibility loci associated with T2D, including over 1200 independent associations of single-nucleotide variants, and approximately 6% of these loci have been successfully repeated within the East Asian population [9,10]. Known environmental and lifestyle factors that interact with genetically determined functions (such as defective insulin secretion and response), such as poor diet (low in fiber), insufficient physical activity, short or disturbed sleep, smoking, stress, depression, and low socioeconomic status, have been associated with a greater risk of T2D [11,12]. Nevertheless, prior investigations into the etiology of T2D have certain limitations. Despite the notable familial clustering of T2D, recognized genetic variations can elucidate only a restricted fraction of individual variances [13]. Gene‒environment interaction analyses have been employed to investigate the joint impact of genetic and environmental risk factors on T2D. However, this approach is still insufficient to account for the unexplained heritability of T2D [14,15]. It is widely acknowledged that environmental and behavioral factors can interact among individuals in close relationships, and many environmental exposures are under the influence of genetic factors [16]. Hence, there is a need for novel family-based approaches to comprehensively investigate the genetic underpinnings of T2D.

In 1998, Jason Wolf et al. reported that parental genotypes can autonomously influence the phenotypes of offspring, irrespective of the transmission of specific genotypes [17]. This phenomenon was introduced as indirect genetic effect (IGEs), signifying the impact of an individual’s genotype on the phenotypes of others through nontransmission pathways [18]. Specifically, within the parent‒child relationship, parents furnish the environment in which their offspring grow and influence their behavior through a much broader set of phenotypes of parents [19]. For example, the educational attainment (EA) of parents having a genetic component provides an environmental effect for children [20]. Consequently, the intergenerational transmission of complex phenotypes is likely attributed to not only genotype transmission but also IGEs. Kong et al. coined the term “genetic nurture effect” to describe the IGEs of parents on their offspring [19]. In recent years, an increasing number of family-based GWASs have been conducted to uncover the IGEs associated with complex phenotypes, such as EA and childhood externalizing disorders [21,22]. Such between-family effects may partially reflect the effects of parental behaviors as well as other aspects of the environment, such as peers or schools. These estimates are informative of the net importance of shared environments because they do not require that the relevant factors are known and measured. In terms of the environmental factors influencing T2D and its risk factors, various aspects of the family environment are hypothesized to play crucial roles throughout developmental stages. For example, 32% of the association between parental history and T2D could be ascribed to environmental effects that are shared among first-degree individuals, and shared environmental effects between twins accounted for 15% of the liability for T2D [23,24]. Owing to the multifactorial nature of T2D and the genetic underpinning underlying its various environmental and lifestyle factors, T2D may be influenced by parental genotypes that are indirectly transmitted to offspring. This effect may occur through IGEs, in addition to the effects of transmitted genetic factors.

GWASs aim to estimate the direct effects of an individual’s own genotype on their own phenotype. However, it has been recognized that general GWAS effect size estimates are confounded by IGEs [17]. Single nucleotide polymorphism (SNP) effects are obtained through general GWASs, referred to here as total genetic effects (TGEs), or population effects, including the causal effects of alleles transmitted by parents or alleles carried by the proband on the traits of interest, called direct genetic effects (DGEs); IGEs; and other confounding effects due to population stratification and assortative mating [25]. Importantly, considering that IGEs are generated by parents nurturing their offspring irrespective of genotype transmission, it can be posited that, for a given genetic variant, the IGEs are equivalent in both the TGEs and the effects of nontransmitted alleles. The effects of nontransmitted alleles essentially capture IGEs, which can be derived from family-based GWASs. Conversely, by excluding IGEs from TGEs, we can isolate direct genetic effects (DGEs), which denote the effects of genes on an individual’s phenotype through direct biological pathways. In addition, recent research has revealed that IGEs associated with certain phenotypes exhibit parent-of-origin effects (PoOs), and T2D reportedly has different transmission effects between fathers and mothers according to observational studies [26]. Decomposing the TGEs estimated by GWASs into distinct components and then dissecting IGEs into maternal and paternal parts provides insights into the diversity of genetic effects, a crucial aspect for identifying molecular mechanisms on the basis of biological interpretation of results [27,28,29].

Many efforts have been undertaken to estimate genetic nurture (indirect) and direct effects on mental diseases or behavioral phenotypes, but there has been little research on T2D, despite some researchers identifying the transmission of parental effects to offspring with T2D in observational studies [27,28]. To fill these knowledge gaps, we firstly conducted SNP-level association analyses and gene-level association analyses on the basis of the Chinese trio family and pinpointed significant loci associated with T2D for TGEs, DGEs, IGEs, and indirect genetic effects transmitted by mothers (IGEs-M) and fathers (IGEs-P). Furthermore, pathway-based association analyses and tissue enrichment analyses were employed to better understand the associated pathways and potential target organs related to T2D.

## 2. Materials and Methods

### 2.1. Study Design and Participants

The study participants were from the Fangshan/Family-based Ischemic Stroke Study in China (FISSIC) [30]. The FISSIC is a community-based genetic epidemiological study. Using proband-initiated contact method, the FISSIC initially collected the ischemic stroke pedigrees and subsequently extended the scope to include pedigrees for T2D and hypertension in Fangshan, a rural area located southwest of Beijing, China. The details of T2D pedigrees collection have been described previously [31,32]. The residents aged older than 18 years were recruited from June 2005 to August 2017. In this study, the inclusion criteria were listed as follows: (1) the participants from T2D pedigrees; (2) sex, age, or T2D conditions were not missing; (3) at least one parent or sibling, along with proband, had genotyping data; (4) no single-gene genetic disease or cancer. A total of 726 subjects within full siblings and 36 subjects within incomplete trios (father–offspring and mother–offspring pairs) were needed to impute missing parental genotypes, as shown in Figure 1. For more details of pedigrees collection for our study, see Appendix A.

### 2.2. Outcome and Demographic Variable Definitions

T2D was defined according to self-reported records or abnormal glycemic markers (fasting blood glucose (FBG) ≥ 7.0 mmol/L or HbA_1c_ ≥ 6.5%), and the diagnosis was confirmed by endocrinologists.

The demographic information, lifestyle risk factors, and self-reported medical history were collected through face-to-face surveys by trained staff. Physical measurements, including height, weight, waist circumference (WC), systolic blood pressure (SBP), and diastolic blood pressure (DBP), were taken by trained investigators. Body mass index (BMI) was calculated as weight (kg) divided by height squared (m^2^), and waist-to-hip ratio (WHR) was calculated as calculated as waist circumference by hip circumference. Laboratory tests of serum lipid levels, including triglyceride (TG), total cholesterol (TC), high-density lipoprotein cholesterol (HDL-C), low-density lipoprotein cholesterol (LDL-C), apolipoprotein A1 (ApoA), and apolipoprotein B (ApoB), were performed by qualified technicians from the laboratory of molecular epidemiology in the department of epidemiology at Peking University. Hypertension (HTN) was defined as a self-reported history and/or systolic blood pressure ≥140 mmHg, diastolic blood pressure ≥90 mmHg, and/or the use of antihypertensive medications.

### 2.3. Genotyping

The whole-blood samples were collected from selected participants. DNA was genotyped using an Infinium Asian Screening Array-24 v1.0 BeadChip with a LabTurbo 496-Standard System (TAIGEN Bioscience Corporation, Taiwan, China). The concentration (A260 nm) and purity (A260 nm/A280 nm) of the DNA were measured via ultraviolet spectrophotometry. They were genotyped in 5 batches, grouped by the origin of the samples and with a balanced case‒control mix on each array. The QC of genotype data was then performed on each sample and each call set via PLINK 1.9 software, accessed on 10 March 2022. (https://www.cog-genomics.org/plink/) [33,34]. For more details, see Appendix A.

### 2.4. Genotype Imputation and Parental Genotype Imputation

This study conducted two stages of imputation. In the first stage, the genotype was first imputed to 1000 Genome Phase 3 reference panels via IMPUTE2 [35,36]. After the first-stage imputation, genetic variants were removed if the imputation info score was <0.70, the sample call rate was <95%, or the minimum allele frequency (MAF) was <1%. A total of 5,187,963 SNPs were available for analysis.

To improve the statistical power of IGEs estimation, we increased the sample size of trios by conducting the second stage of imputation via the single-nucleotide imputation of parents (snipar) tool, a Python package run in Python v3.9 [25]. Snipar can impute missing parental genotypes based on genotype information of full-sibling or incomplete trio (parent–offspring pairs), detailed in Appendix A. The genetic information of 735 offspring and their parents can be used to estimate IGEs, resulting in a total of 881 offspring when combining complete trios (Figure 1).

### 2.5. Statistical Analysis

Data were analyzed from July 2023 to March 2024, describing the characteristics of offspring in families as mean (SD) for continuous variables and percentages for categorical variables, grouped by T2D diagnosis. Wilcoxon rank sum tests were used for continuous variables and Pearson’s chi-squared tests were applied to categorical variables.

All variants were tested for associations with T2D, assuming an additive model of inheritance within a regression framework, including age and sex. To account for population structure and relatedness, association analyses were performed via a linear mixed model (LMM) with a kinship matrix implemented [37]. This approach produces TGEs estimations of SNPs. Next, we estimated the different type of genetic effect by decomposing the TGEs into DGEs, IGEs, and, more specifically, IGEs-M and IGEs-P. IGEs-M and IGEs-P are the indirect genetic effects that represent the effects of nontransmitted alleles by the mother and father on the offspring phenotypes, respectively. All analyses were performed using the snipar software [25]. Sensitivity analysis was conducted to examine the robustness of the main results. We determined the association between significant SNPs and T2D using four LMM models with different covariates: a model with sex and age groups based on tertiles; a model including sex, age, and age squared terms; a model with sex, age, and top two principal components; and a model with sex, age, and top five principal components. Principal components of genotype data could reveal any possible population stratification.

Functional annotations of the each SNPs were performed using FUMA, a web-based tool to facilitate functional mapping of SNP analysis results (https://fuma.ctglab.nl (accessed on 20 October 2023)) [38]. We utilized the SNP2GENE function within the FUMA to identify significant genomic loci of different genetic effect type. A genomic locus was defined as an about 500 kb long genomic region that includes all SNPs in LD (r^2^ ≥ 0.6), having at least one lead SNP. The lead SNP achieves the significant threshold and is the top significant SNP of a genomic locus. In addition, we assigned the significant SNPs within each genomic risk locus to their nearest genes (also called mapped genes) based on genomic position and functional annotations using FUMA, including the combined annotation dependent depletion (CADD) score [39], the probability of regulatory functionality (RegulomeDB score) [40], and transcriptional/regulatory effects derived from chromatin states (the minimum chromatin state) [41]. For more details of genomic locus identification and SNP mapping, see Appendix A.

The SNP-based *p* values of each type of genetic effect were used for gene-based analysis via MAGMA software (v1.10), a useful tool for gene and gene-set analysis [42]. A total of 18,463 protein-coding genes from the database (NCBI 37.3) were used for SNP mapping. Under the multiple regression approach, MAGMA properly incorporates LD between SNPs and detects multiple SNP effects for genome-wide gene association analysis. We applied a stringent Bonferroni correction to account for multiple tests.

A competitive pathway-based analysis was performed via the GSA-SNP2 tool (released 1 September 2020) [43]. Pathway-based analysis substantially improved the statistical power of GWASs because it can detect aggregate associations of multiple SNPs or genes even when the individual candidates are only mildly associated. Many novel genes and pathways not found via the SNP-based approach can be identified for better biological interpretations. For pathways annotation, we selected BioCarta, Pathway Interaction (PID), Kyoto Encyclopedia of Genes and Genomes (KEGG), Reactome, and WikiPathways databases. The ontologies were derived from the Gene Ontology (GO) resource containing biological process (BP), cellular component (CC), and molecular function (MF) components and the Human Phenotype Ontology (HPO) databases. This annotation information has been integrated into a dataset from the Human Molecular Signatures Database (MSigDB) [44]. During the analyses, the parameters are set to their defaults, as in previous publications [43]. More details of analysis can be found in Appendix A. The Benjamini–Hochberg method was used for the multiple testing correction. Associated ontologies/pathways were selected only if the adjusted *q* value was <0.05.

To better understand the potential target organs of different types of genetic effects on T2D, tissue enrichment analysis was performed via the R package TissueEnrich (v1.24.1) on the candidate genes identified via SNP-based, gene-based, or pathway-based approaches [45]. Processed RNA-seq data from the Human Protein Atlas (HPA, https://www.proteinatlas.org (accessed on 10 February 2024)) database were used to define tissue-specific genes. We used tissues with more than 1 biological replicate for more robust calculations, resulting in 32 of tissues in the HPA. We then applied a Benjamini–Hochberg correction to all gene set associations to account for multiple testing. Associated gene sets were selected only if adjusted *p* value < 0.05.

## 3. Results

### 3.1. Description of the Study Population

Descriptive statistics for the key study variables are presented in Table 1. The case group consisted of a total of 291 offspring with T2D, with an average age of 57.94 ± 9.10 years. The control group comprised 590 offspring without T2D, with an average age of 54.81 ± 10.91 years. The sex distributions in both groups were similar, with a slightly greater proportion of males than females. Significant differences were observed between the case and control groups in terms of age and the prevalence of HTN, WC, WHR, FBG, HbA1c, and TG levels (*p* < 0.05). The baseline characteristics of their parents (if observed) are shown in Appendix A in Appendix A.

### 3.2. SNP-Based Analyses Decompose the Genetic Effects on Offspring with Type 2 Diabetes

Figure 2a–e and Table 2 summarize the results from the SNP-based association analyses of each type of genetic effect. We identified four SNPs (two loci) that met a genome-wide significance threshold of *p* < 5 × 10^−8^,three of which were identified in DGEs (Figure 2b) and one of which was identified in IGEs-P (Figure 2e). The SNPs rs3866325 (*p* < 2.59 × 10^−8^), rs202048780 (*p* < 2.39 × 10^−8^), and rs9861368 (*p* < 4.42 × 10^−8^) were negatively associated with T2D in DGEs, reflecting the direct genetic association. They were located in the same genomic locus and the lead SNP is rs202048780. The top association signal at this locus was intergenic near long intergenic nonprotein coding RNA 879 (*LINC00879*[HGNC 48566]) in chromosome 3. In addition, the lead SNP rs3805116 (*p* < 4.39 × 10^−8^), located in the intron of the mitochondrial ribosomal protein S33 gene (*MRPS33* [OMIM 611993]) gene in chromosome 7, was significant in IGEs-P, reflecting the indirect genetic association from the father. The overall performance was robust in sensitivity analysis, with the effect size of four SNPs changed among 0%~55% relative to the main result (Appendix A). In addition, the suggestively significant SNPs at the 5 × 10^−8^ < *p* < 5 × 10^−6^ threshold, including 379 signals and 54 mapped genes, are detailed in Appendix A (Appendix A).

### 3.3. Gene-Based Analyses Enhance the Discovery of Novel Associated Genes from DGEs and IGEs GWASs

To improve the power of T2D-associated gene discovery, we grouped SNPs within genes and calculated their aggregated association on T2D (Table 3). In the gene-based analysis, we identified five significant associations of genes with T2D at *p* < 2.71 × 10^−6^ (0.05/18,467) (Figure 3). Among these genes, PACRG was positively associated with T2D in TGEs (*p* = 2.12 × 10^−6^) and DGEs (*p* = 1.29 × 10^−6^). The *MRPS33* (*p* = 1.58 × 10^−6^), *PIH1D2* (*p* < 6.76 × 10^−7^), and SDHD (*p* < 2.67 × 10^−6^) genes were positively associated with T2D in IGEs-P.

### 3.4. Identification of the Associated Pathways for Better Biological Interpretation

Gene set enrichment analyses were conducted via the summary statistics of three GWASs of T2D with different type of genetic association. A total of 56, 41, 20, 0, and 4 terms passed the FDR correction in ontologies, whereas 18, 34, 1, 0, and 2 pathways showed significant associations with T2D in TGEs, DGEs, IGEs, IGEs-M, and IGEs-P, respectively (Figure 4a). In the ontology enrichment analysis, five ontologies were significantly enriched in both DGEs and IGEs (adjusted *q* value < 0.05), including flavonoid metabolic process, entropion and antigen processing and presentation via MHC class Ib, cellular glucuronidation, and postsynaptic density membrane, and one term, called response to cocaine, was enriched in both DGEs and IGEs-P (Figure 4a). Notably, three ontologies were significantly enriched only in IGEs-P: the regulation of systemic arterial blood pressure and the secretion and transport of catecholamines. In the pathway enrichment analysis, no pathways were enriched in either DGEs or IGEs (Figure 4b). Two pathways were significantly enriched in the IGEs-P group, including the transforming growth factor beta (TGF-β) pathway (genes upregulated during epithelial‒mesenchymal transition upon TGF-β stimulation) and the TGF-β pathway (genes upregulated upon TGF-β stimulation). The results are presented in the Appendix A (Appendix A).

### 3.5. Tissue-Specific Enrichment of Important Target Organs of Candidate Genes

We used TissueEnrich tools to find tissues enriched with genes identified via SNP-based, gene-based, or pathway-based analyses. Respectively, 14, 10, 2, and 4 tissues were significantly enriched in TGEs, DGEs, IGEs, and IGEs-P within 32 types of tissue within the HPA database (Figure 5), whereas 16, 9, 6, and 1 tissues were significantly enriched in TGEs, DGEs, IGEs, and IGEs-P. Specifically, the genes in DGEs and IGEs were both significantly enriched in liver tissue and cerebral cortex tissue. Moreover, we also found that genes identified by DGEs or IGEs-P were significantly enriched in distinct tissues: the enrichment of gallbladder, smooth muscle, and adrenal gland tissues was observed only in IGEs-P, whereas the enrichment of lymph node, kidney, spleen, small intestine, duodenum, bone marrow, appendix, and tonsil tissues was observed only in DGEs, as detailed in Appendix A (Appendix A).

## 4. Discussion

In this study, we identified two novel loci (two genes) associated with T2D risk in the Chinese population through dissecting TGEs into DGEs and IGEs at the SNP level. Four genes were positively associated with T2D, and of those, one locus was successfully replicated in gene-based analysis. Through pathway-based analysis, we observed that some ontologies and pathways were significantly enriched among different types of genetic effect. For tissue enrichment analysis, liver and cerebral cortex were identified as significantly enriched tissues in both DGEs and IGEs, although few overlapping genes were identified via the SNP-based, gene-based, and pathway-based approaches. Moreover, we also found that gallbladder, smooth muscle, and adrenal gland tissues were significantly enriched only in IGEs, revealing shared and distinct molecular mechanisms underlying the direct and indirect genetic effects on T2D in the Chinese population.

The results of SNP-based analysis revealed that three variants, located near the long noncoding RNA *LINC00879*, may decrease the genetic susceptibility to T2D. Long noncoding RNA (lncRNA) expression is typically characterized by low abundance but greater tissue and cell specificity [46]. Previous studies have indicated that the islet properties of lncRNAs are approximately five times greater than those of common protein-coding genes [47]. Research on the relationship between lncRNAs and metabolic diseases remains limited, suggesting a potential association between these genes and diabetes. Notably, *LINC00879* was previously linked to alcohol consumption per week in a multiethnic GWAS [48]. Alcohol consumption is recognized as a significant risk factor for T2D, implying that *LINC00879* may exhibit gene pleiotropy, affecting both islet function and drinking behavior. However, despite detecting significant associations between the DGEs of SNPs near *LINC00879* and T2D, the TGEs did not surpass the statistical threshold set by the GWAS, and the IGEs moved in the opposite direction to the DGEs. Consequently, its impact on T2D may be masked by its adverse effects on drinking behavior. These findings underscore the importance of deconstructing population effects into DGEs and IGEs, as they may aid in mitigating confounding stemming from gene pleiotropy.

We found a locus having positive IGEs on T2D predisposition, with its lead SNP being rs3805116. The locus, mapped to the *MRPS33* gene, was subsequently successfully repeated via gene-based analysis. *MRPS33* is a protein-coding gene responsible for encoding mammalian mitochondrial ribosomal proteins. Interestingly, *MRPS33* has previously been associated with schizophrenia [49]. Schizophrenia often presents with comorbidities such as T2D, and the prevalence of T2D is greater in individuals with a family history of mental illness [50]. Extensive research has highlighted the significant role of genes, including genes such as *5-HTR2C*, *CHRNA*, *IGF2BP2*, and TPH, in the susceptibility of patients with schizophrenia to comorbid T2D [51]. Compared with those of previous studies, the findings of our study shed further light on how the genetic basis of schizophrenia can not only impact an individual’s mental health and susceptibility to T2D but also influence T2D risk in the next generation, even when the risk alleles are not directly transmitted to their offspring. These insights suggest that future strategies for preventing complex diseases may be more effectively targeted at the family level than solely at the individual level.

Gene-based analyses revealed that the *PACRG* gene exhibited significantly positive DGEs, as did the TGEs, on T2D. *PACRG* was previously identified through GWASs as a gene linked to blood lipid levels in individuals of European descent and subsequently validated in the Chinese population [52,53]. Dyslipidemia is a pivotal physiological mechanism associated with T2D, and it is plausible that this may represent the biological pathway through which *PACRG* manifests its DGEs. Furthermore, we detected significant IGEs-P in T2D associated with the genes *PIH1D2* and *SDHD*. Notably, there are few reports regarding these two genes and T2D-related traits in prior studies.

This is the first study demonstrating the IGEs on T2D susceptibility, revealing the importance of family nurturing in decreasing T2D risk. In gene-set analysis, the flavonoid metabolic process was identified as being positively associated with T2D in both DGEs and IGEs, suggesting that flavonoid metabolism was partly affected by individual genetics but also the parental nurturing effects may provide a synergistic effect to enhance the risk of T2D. Flavonoids are naturally occurring polyphenols that are commonly consumed through diets, and it has been reported that these compounds can safely and efficiently prevent and cure T2D and its complications [54,55]. The results lay the foundation for the development of novel family-based diet prevention strategies for T2D. We also revealed that cellular glucuronidation and immune-related and neurotransmitter transmission-related ontologies were positively associated with T2D. These findings have been reported in previous studies on T2D genetic mechanisms, but proper family nurturing may modify an individual’s high genetic predisposition on T2D based on our results [56,57,58].

Interestingly, the IGEs of SNPs, genes, and pathways on T2D may be partly attributed to fathers, as our results revealed an potential paternal effect of T2D in the family nurture environment. The function of IGEs-P involves the regulation of systemic arterial blood pressure, as well as the secretion and transport of catecholamines and the TGF-β-related pathways. Few studies have explored the effect of paternal nurture (paternal phenotypes or behaviors) on T2D risk. In contract, many studies have found that environmental factors in utero were associated with a higher risk of developing T2D in offspring, suggesting that the intrauterine environment during pregnancy may be one of the mediators mediating the relationship between maternal genotype and offspring phenotype [59,60]. However, publications on exploring intrauterine environment mediating IGEs are limited. In addition, our study did not identify maternal indirect genetic effects on T2D, although Wang et al. and Bener et al. reported that maternal effects play a significant role in the transmission of T2D in observational studies [27,28]. Differences in longevity between men and women, ethnic groups, and eating habits in families, and the complexity of sources of maternal effects, could lead to differing conclusions about parental effects. More effort should be made to consider incorporating more type of pedigrees, such as adoptive families, to elucidate the mediator in IGEs, such as intrauterine effects, similar to the work conducted by Hwang et al. on EA [61].

According to tissue enrichment analysis, we observed enrichment of cerebral cortex regardless of the type of genetic effect, suggesting that the intrinsic neural function mechanism is involved in both DGEs and IGEs. The former is influenced by family nurturing, whereas the latter is attributed to individual heredity. Research has revealed that T2D risk genes are associated with neuromodulation-related functions [9,62]. No publication has explored the molecular mechanisms of DGEs and IGEs on T2D. Our work addresses this gap to an extent, showing that some tissues are significantly enriched only in DGEs or IGEs. For example, the enrichment of gallbladder, smooth muscle, and adrenal gland tissues were found only in IGEs-P, hinting that the father’s diet or activity may be associated with T2D in offspring.

Several limitations must be recognized. All participants in this study were of East Asian ancestry, which does not account for the varying genetic backgrounds present in trans-ancestry groups. More loci are expected to be discovered as the sample size increases or as other ancestries are considered. Moreover, as IGEs occur within closely related individuals, the effect size is anticipated to be influenced by variables such as family economic conditions and the degree of intimacy between parents and offspring. This type of bias cannot be corrected with the LMM developed by sniper [25]. More effort should be made for methodological improvement to account for those variables.

## 5. Conclusions

Our findings provide a novel perspective on the role of parent nurturing in the prevention of T2D. We identified the significant paternal indirect genetic association for several loci and genes, including *MRPS33,* suggesting the fathers’ nurturing role in T2D risk. Paternal effects on T2D were involved with pathways of blood pressure regulation and neurotransmitter pathways. Thus, this study highlights the importance of paternal nurturing behavior for reducing T2D susceptibility in offspring, facilitating the development of individualized family-based prevention strategies.

## Figures and Tables

**Figure 1 biomedicines-13-00120-f001:**
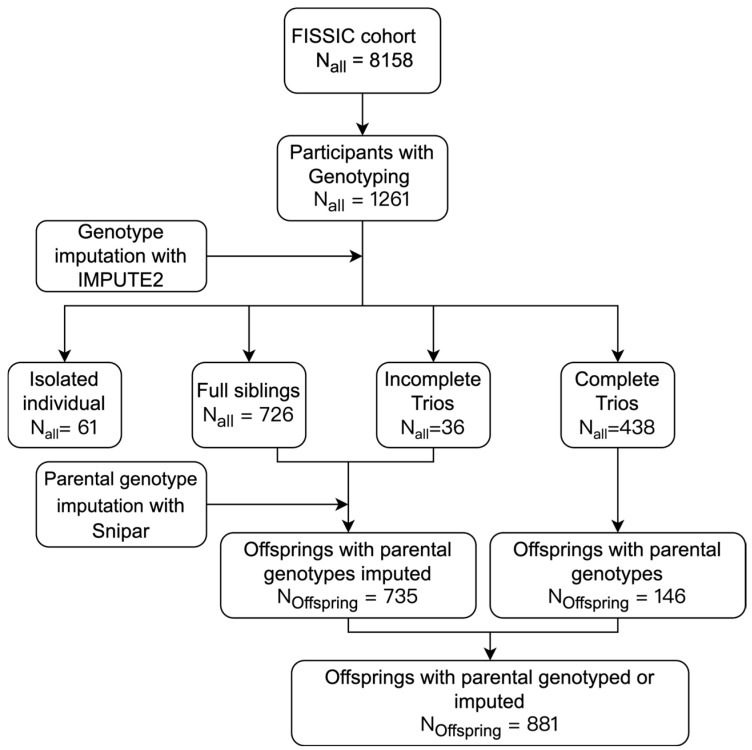
Flow diagram of participants selection in this study. Based on the diagnoses confirmed by endocrinologists, 1261 participants having genotype data were selected from the FISSIC cohort. After genotype imputation, we excluded isolated individuals because their relatives were removed after quality control (QC) procedures. For full siblings and incomplete trios, parental genotype inference was performed to increase the number of trios for better estimation of IGEs.

**Figure 2 biomedicines-13-00120-f002:**
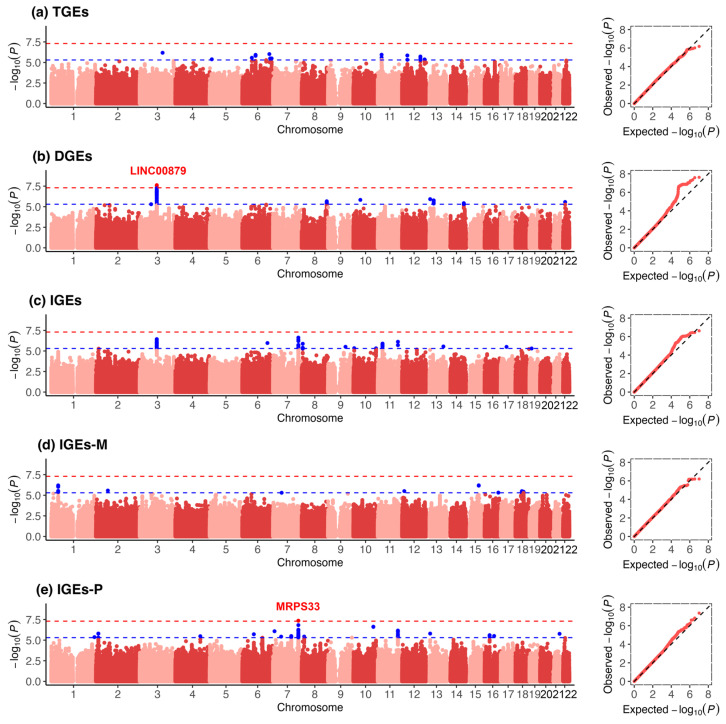
Manhattan plots and quantile‒quantile plots of different genetic effects on T2D at SNP level for (**a**) Total genetic effects (TGEs); (**b**) Direct genetic effects (DGEs); (**c**) Indirect genetic effects (IGEs); (**d**) Maternal indirect genetic effects (IGEs-M); (**e**) Paternal indirect genetic effects (IGEs-P). In the left plots of each subfigure, the top variants in genomic loci are labeled in the plots. The plot on the left shows the negative log10-transformed *p* values concerning the significant SNPs and suggestively significant SNPs (rsID) and their chromosomal locations (Hg19). The red and blue horizontal dashed lines indicate the negative log10-transformed *p* thresholds 5 × 10^−8^ and 5 × 10^−6^, respectively. The labels represent the mapped genes of the loci with lead SNP *p* < 5 × 10^−8^. The plot on the right shows the theoretical distribution of *p* values under a uniform distribution (*x*-axis) versus the observed distribution within the sample (*y*-axis).

**Figure 3 biomedicines-13-00120-f003:**
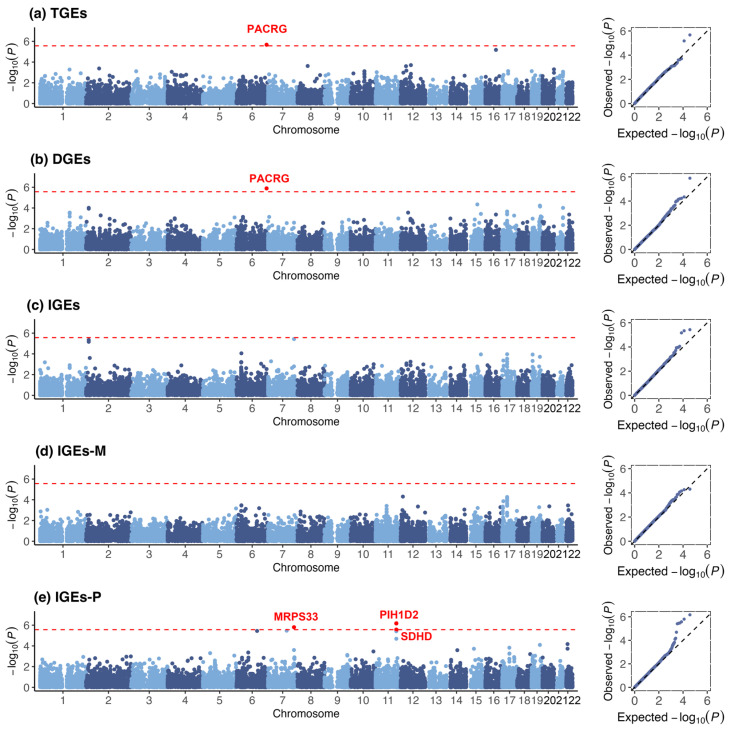
Manhattan plots and quantile‒quantile plots of different genetic effects on T2D at gene level for (**a**) Total genetic effects (TGEs); (**b**) Direct genetic effects (DGEs); (**c**) Indirect genetic effects (IGEs); (**d**) Maternal indirect genetic effects (IGEs-M); (**e**) Paternal indirect genetic effects (IGEs-P)**.** In the left plots of each subfigure, the plot on the left shows the negative log10-transformed *p* values with respect to the significant genes and their chromosomal locations (Hg19). The red horizontal dashed lines indicate that the negative log10-transformed *p* threshold is 2.71 × 10^−6^. The labels represent the significant genes. The plot on the right shows the theoretical distribution of *p* values under a uniform distribution (*x*-axis) versus the observed distribution within the sample (*y*-axis).

**Figure 4 biomedicines-13-00120-f004:**
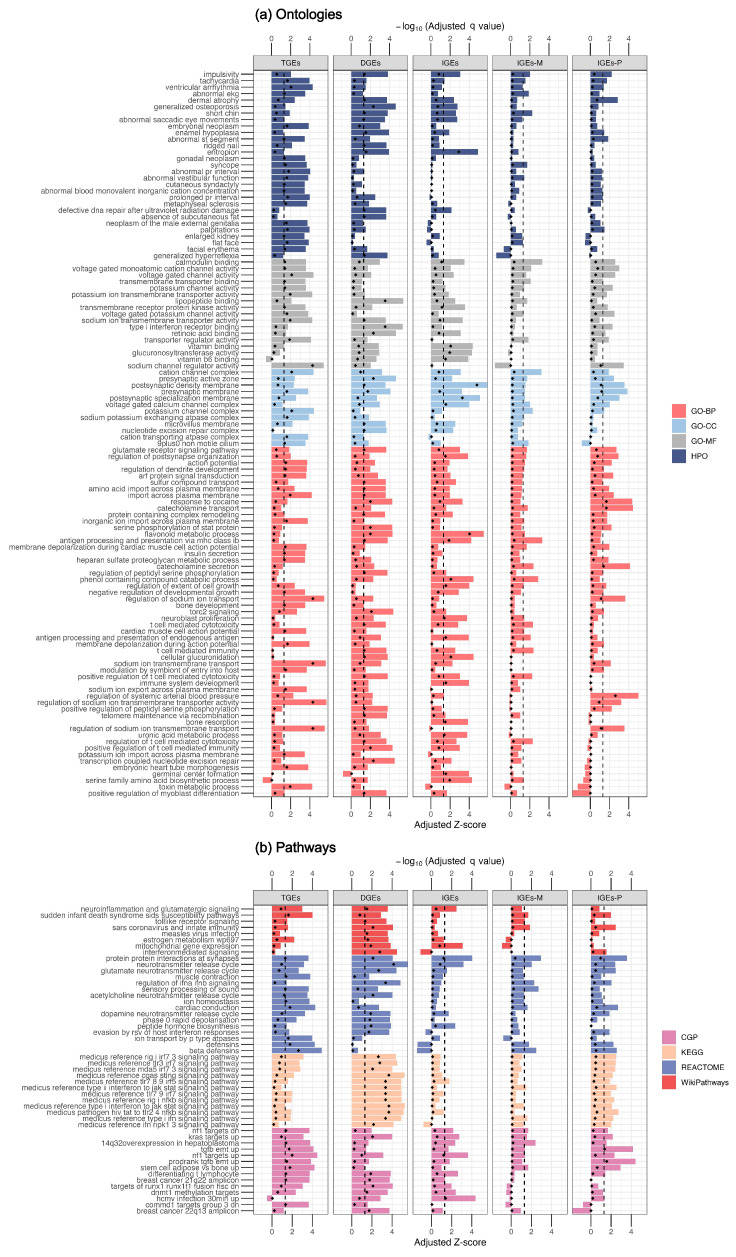
Pathway-based analysis results of different genetic effects on T2D. (**a**,**b**) are the horizontal bar graphs representing the results based on ontologies and pathways annotations of the gene sets, respectively. Negative log10-transformed adjusted *q* values and adjusted *z* scores (*z-*score evaluated with adjusted gene score) are plotted on a double *y*-axis with respect to the ontologies and pathways(*x*-axis). The black vertical dashed line indicates the negative log10-transformed adjusted q threshold of 0.05. The bar length represents the effect size of gene-sets (the adjusted *z* score), and the position of the points represents the size of the adjusted *q* values.

**Figure 5 biomedicines-13-00120-f005:**
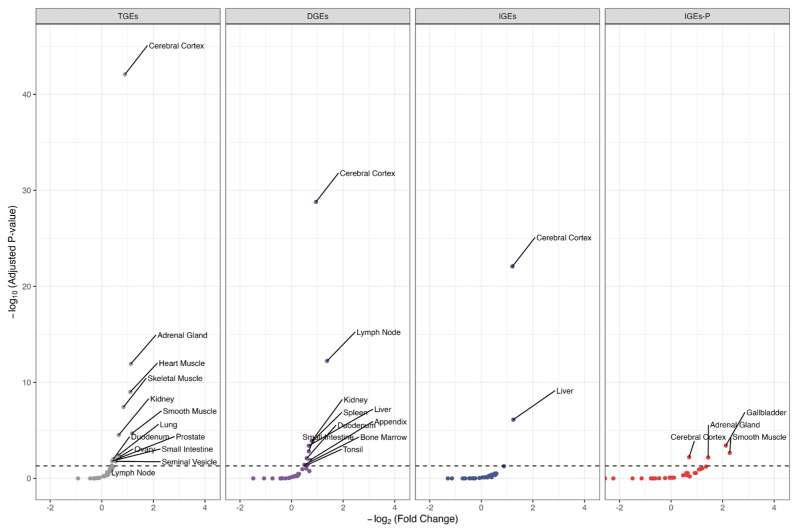
Tissue enrichment results of candidate genes (HPA database). Negative log10-transformed adjusted *p* values are plotted against the fold change (effect size) based on the ontologies and pathways. The black dashed horizontal line indicates the threshold for negative log10-transformed adjusted *p* values of 0.05.

**Table 1 biomedicines-13-00120-t001:** Baseline characteristics of the offspring group (mean ± SD or *n* (%)).

Characteristics	Overall *n* = 881	No T2D *n* = 590	T2D *n* = 291	*p* Value ^1^
Sex, *n* (%)				0.375
Female	363 (41.20%)	237 (40.17%)	126 (43.30%)	
Male	518 (58.80%)	353 (59.83%)	165 (56.70%)	
Age, years	55.84 ± 10.44	54.81 ± 10.91	57.94 ± 9.10	<0.001
HTN, *n* (%)	621 (70.57%)	388 (65.87%)	233 (80.07%)	<0.001
Education, *n* (%)				0.078
Illiteracy	150 (17.38%)	96 (16.61%)	54 (18.95%)	
Primary school	164 (19%)	97 (16.78%)	67 (23.51%)	
Middle school	387 (44.84%)	267 (46.19%)	120 (42.11%)	
High school	130 (15.06%)	95 (16.44%)	35 (12.28%)	
University	32 (3.71%)	23 (3.98%)	9 (3.16%)	
Smoke, *n* (%)	455 (52.66%)	311 (53.81%)	144 (50.35%)	0.340
Drink, *n* (%)	375 (43.50%)	257 (44.46%)	118 (41.55%)	0.420
Height, cm	163.23 ± 8.01	163.50 ± 8.12	162.69 ± 7.77	0.230
WC, cm	91.34 ± 9.38	90.75 ± 9.48	92.54 ± 9.07	0.008
BMI, kg/m^2^	26.26 ± 3.50	26.12 ± 3.56	26.53 ± 3.36	0.120
WHR, cm/cm	0.91 ± 0.06	0.91 ± 0.06	0.93 ± 0.06	<0.001
SBP, mmHg	138.49 ± 20.25	137.67 ± 20.12	140.16 ± 20.45	0.100
DBP, mmHg	83.72 ± 11.57	83.88 ± 11.63	83.40 ± 11.47	0.380
FBG, mmol/L	5.58 ± 2.43	4.64 ± 0.91	7.48 ± 3.29	<0.001
HbA_1c_, %	6.51 ± 1.46	5.78 ± 0.30	7.70 ± 1.78	<0.001
TG, mmol/L	1.55 ± 1.38	1.47 ± 1.31	1.70 ± 1.50	0.002
TC, mmol/L	3.11 ± 0.89	3.09 ± 0.86	3.14 ± 0.93	0.600
HDL-C, mmol/L	0.93 ± 0.31	0.94 ± 0.30	0.90 ± 0.31	0.040
LDL-C, mmol/L	2.19 ± 0.80	2.18 ± 0.75	2.22 ± 0.90	0.800
ApoA, mmol/L	1.12 ± 0.31	1.13 ± 0.29	1.12 ± 0.34	0.680
ApoB, mmol/L	0.74 ± 0.23	0.73 ± 0.22	0.76 ± 0.24	0.210

^1^ Pearson’s chi-squared test; Wilcoxon rank sum test.

**Table 2 biomedicines-13-00120-t002:** Significant SNPs from family-based GWASs of different genetic effects.

CHR	SNP	POS	Mapped Gene ^1^	A1/A2	Freq	Effect Type	Beta	SE	*p* Value
3	rs3866325	94749380	*LINC00879*	T/A	0.303	TGEs	−0.09	0.03	6.17 × 10^−4^
						DGEs	−0.22	0.04	2.59 × 10^−8^
						IGEs	0.13	0.03	3.68 × 10^−7^
						IGEs-M	0.22	0.05	3.01 × 10^−5^
						IGEs-P	0.05	0.05	3.23 × 10^−1^
3	rs202048780	94750315	*LINC00879*	ATAG/-	0.299	TGEs	−0.09	0.03	5.74 × 10^−4^
						DGEs	−0.22	0.04	2.39 × 10^−8^
						IGEs	0.14	0.03	3.63 × 10^−7^
						IGEs-M	0.20	0.05	9.59 × 10^−5^
						IGEs-P	0.07	0.05	1.80 × 10^−1^
3	rs9861368	94756030	*LINC00879*	T/G	0.304	TGEs	−0.09	0.03	8.66 × 10^−4^
						DGEs	−0.22	0.04	4.42 × 10^−8^
						IGEs	0.13	0.03	3.92 × 10^−7^
						IGEs-M	0.20	0.05	6.28 × 10^−5^
						IGEs-P	0.07	0.05	2.06 × 10^−1^
7	rs3805116	140713367	*MRPS33*	C/T	0.082	TGEs	−0.02	0.04	6.91 × 10^−1^
						DGEs	−0.23	0.07	3.99 × 10^−4^
						IGEs	0.22	0.04	2.37 × 10^−7^
						IGEs-M	−0.11	0.09	2.63 × 10^−1^
						IGEs-P	0.54	0.10	4.39 × 10^−8^

^1^ Mapped gene: defined by FUMA (https://fuma.ctglab.nl (accessed on 20 October 2023)).

**Table 3 biomedicines-13-00120-t003:** Genes significantly associated with T2D in different genetic effects.

GENE	CHR	POS	NSNPs ^1^	Effect Type	ZSTAT	*p* Value
*PACRG*	6	163148164–163736524	1112	TGEs	4.60	2.12 × 10^−6^
				DGEs	4.70	1.29 × 10^−6^
				IGEs	2.76	3.00 × 10^−3^
				IGEs-M	1.85	3.24 × 10^−2^
				IGEs-P	0.28	3.88 × 10^−1^
*MRPS33*	7	140705854–140715028	13	TGEs	−2.73	9.97 × 10^−1^
				DGEs	2.79	3.00 × 10^−3^
				IGEs	4.48	3.71 × 10^−6^
				IGEs-M	−0.37	6.43 × 10^−1^
				IGEs-P	4.66	1.58 × 10^−6^
*PIH1D2*	11	111934734–111944998	2	TGEs	0.19	4.24 × 10^−1^
				DGEs	1.16	1.22 × 10^−1^
				IGEs	1.14	1.27 × 10^−1^
				IGEs-M	2.46	7.00 × 10^−3^
				IGEs-P	4.83	6.76 × 10^−7^
*SDHD*	11	111957497–111990353	24	TGEs	0.14	4.44 × 10^−1^
				DGEs	1.28	1.00 × 10^−1^
				IGEs	1.20	1.16 × 10^−1^
				IGEs-M	2.62	4.00 × 10^−3^
				IGEs-P	4.55	2.67 × 10^−6^

^1^ The numbers of SNPs in the defined genomic loci.

## Data Availability

The datasets examined in this study can be obtained from the corresponding author upon reasonable request.

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
