# Peer review of "Genetic Nurture Effects on Type 2 Diabetes Among Chinese Han Adults: A Family-Based Design"

_biomedicines, 2025, doi:10.3390/biomedicines13010120_

Round 1
Reviewer 1 Report
Comments and Suggestions for Authors
Dear correspondence author, it was very nice reviewing your research work. The research study illustrates fundamental recommendations and outlined important paternal genetic factors. However, studies wouldn’t be completed without facing some changes and challenges. Kindly, follow below the suggested notes and wish you all the best.
General comments:
- Pay attention to the square brackets of the references in the text, some they appear after the dot at the end of the sentence.
- Legend in figures 1 should include extra information and few more sentences showing the importance of the data represented above the figure.
- Carefully use the uppercase letter, for example line 30, 145, 146. Kindly adjust them.
- In genotyping section: Kindly define the absorbance wavelength used in both methods (the purity and concertation measurements).
- Give a full scientific name-description when listing the abbreviation for the first time. For example, (PCA), line number 158. Also, linear mixed models (LMM) line 439.
- Line 234: Add this information (www.proteinatlas.org) after The Human Protein Atlas.
- In addition, make it consistence for other capitalization letter to lowercase around the whole text, for example line 225-230, 234 and 259.
- Define WC abbreviation either in line 141 or 247. Furthermore, the value of WC is missing in table 1.
- Adjust the quality and enhance the resolution of the figures (Figure 2 and 3), none of the red and green dashed lines clearly seen. However, in fig.3 only one dashed line presented. Use the conversion metafile from data on excel to word document. This will improve the quality of the components in the images while in transferred to the word document.
- Figure 2 and 3 (a-e), how about Chr20 and 22, are not shown in the x-axis.
- In table 2, correct the A1/A2 of the rs202048780 (TATAG/T).
- All the best with your publication.
Author Response
We would like to express our sincere gratitude to you for the comments and suggestions on our article. All of the comments are detailed and constructive for revising and improving our manuscript, as well as the important guiding significance to our research. The detailed corrections are listed below. We also submit a revised version of the manuscript and the corrections to our manuscript were all highlighted within the document by using blue-colored text in this revised version. Please refer to the attached file for more details.

Reviewer 2 Report
Comments and Suggestions for Authors
This is an interesting manuscript, the genetic information reported is relevant, but genetic information cannot be the sole responsibility for current type 2 diabetes. The suggestion is to direct the information towards a predisposition to diabetes.
Author Response
We would like to express our sincere gratitude to you for the comments and suggestions on our article. The detailed corrections are in attached file. We also submit a revised version of the manuscript and the corrections to our manuscript were all highlighted within the document by using blue-colored text in this revised version. Please refer to the attached file for more details.

Reviewer 3 Report
Comments and Suggestions for Authors
The authors address a very important issue, namely the genetic etiology of type 2 DM including indirect genetic effects. This should make the ms of interest to a wide range of clinicians and epidemiologist. I am afraid however that the overly technical style makes the ms hard to follow for the majority of those people. I have to admit that I also felt a bit out of my depth at times. It would therefore be helpful if the authors would revise the paper in a way that makes it more accessible, e.g. by relegating technical details to supplementary material and focusing on the logic of their reasoning.
I have a few concerns that I hope the authors can address.
First, the data appears to be not population based, but (on the basis of the protocol paper that they reference. There seems to be no publication in English presenting results ?) a sample collected on the basis of the presence of stroke, a disease that may itself be under the control of various genes. The extent to which this affects the validity of methods used is unclear. Perhaps the authors can dwell a bit on this issue and also present more details of the cohort data.
Another concern is the absence (as far as I could see) of parental phenotypical data (ie T2D in parents). If there are indirect genetic effects I would like to know whether the causal mechanism is via the parental phenotypes or not. This is also of interest in the context of possible intrauterine effects (not mentioned).
Another, perhaps minor, point is the treatment of age in their regression. T2D is - independent of genes - strongly age dependent and the age dependency is - if I remember ok- not linear.
Finally, given all the different steps - each with underlying assumptions - some kind of sensitivity analysis (or simulation study) might add value to the ms
Author Response
We would like to express our sincere gratitude to you for the comments and suggestions on our
article. All of the comments are detailed and constructive for revising and improving our
manuscript, as well as the important guiding significance to our research. The detailed
corrections are listed in attached file. We also submit a revised version of the manuscript and the
corrections to our manuscript were all highlighted within the document by using blue-colored
text in this revised version. For more detail, please see attached file.

Round 2
Reviewer 2 Report
Comments and Suggestions for Authors
It is appreciated that the observations were taken into account. In this era of genetic information, we must take care of the attribution of the genetic load, which makes us more susceptible or increases the predisposition to turn on or off genes or regions of the genes involved in diseases. But I believe that the absolute truth is not yet known, it is the epigenetic plus environmental factors plus the genetic load that influence the explanations for the origin of diseases. I agree with this version, which I consider will be of great interest.
Reviewer 3 Report
Comments and Suggestions for Authors
I am impressed by the improvements made by the authors
My first comments were a bit imprecise but the authors interpreted everything perfectly.